# Baricitinib Attenuates Bleomycin-Induced Pulmonary Fibrosis in Mice by Inhibiting TGF-β1 Signaling Pathway

**DOI:** 10.3390/molecules28052195

**Published:** 2023-02-27

**Authors:** Songtao Gu, Jingjing Liang, Jianwei Zhang, Zhichao Liu, Yang Miao, Yuli Wei, Shimeng Li, Jinying Gu, Yunyao Cui, Ting Xiao, Xiaohe Li, Cheng Yang

**Affiliations:** 1Department of Respiratory & Critical Care Medicine, Tianjin Chest Hospital, No. 261 Taierzhuang South Road, Jinnan District, Tianjin 300222, China; 2State Key Laboratory of Medicinal Chemical Biology, College of Pharmacy and Tianjin Key Laboratory of Molecular Drug Research, Nankai University, Haihe Education Park, 38 Tongyan Road, Tianjin 300353, China; 3Tianjin Key Laboratory of Molecular Drug Research, Tianjin International Joint Academy of Biomedicine, Tianjin 300457, China; 4Tianjin Jikun Technology Co., Ltd., Tianjin 301700, China

**Keywords:** pulmonary fibrosis, baricitinib, TGF-β1 signaling pathway, JAK-STAT

## Abstract

Idiopathic pulmonary fibrosis (IPF) is a chronic progressive interstitial lung disease with unknown etiology, high mortality and limited treatment options. It is characterized by myofibroblast proliferation and extensive deposition of extracellular matrix (ECM), which will lead to fibrous proliferation and the destruction of lung structure. Transforming growth factor-β1 (TGF-β1) is widely recognized as a central pathway of pulmonary fibrosis, and the suppression of TGF-β1 or the TGF-β1-regulated signaling pathway may thus offer potential antifibrotic therapies. JAK-STAT is a downstream signaling pathway regulated by TGF-β1. JAK1/2 inhibitor baricitinib is a marketed drug for the treatment of rheumatoid arthritis, but its role in pulmonary fibrosis has not been reported. This study explored the potential effect and mechanism of baricitinib on pulmonary fibrosis in vivo and in vitro. The in vivo studies have shown that baricitinib can effectively attenuate bleomycin (BLM)-induced pulmonary fibrosis, and in vitro studies showed that baricitinib attenuates TGF-β1-induced fibroblast activation and epithelial cell injury by inhibiting TGF-β1/non-Smad and TGF-β1/JAK/STAT signaling pathways, respectively. In conclusion, baricitinib, a JAK1/2 inhibitor, impedes myofibroblast activation and epithelial injury via targeting the TGF-β1 signaling pathway and reduces BLM-induced pulmonary fibrosis in mice.

## 1. Introduction

Pulmonary fibrosis (PF) is a heterogeneous disease caused by many kinds of lung injury, including toxicity, autoimmune, drug or traumatic injuries [1]. The pathological features of PF are over the deposition of extracellular matrix, abnormal repair of damaged epithelial cells and the overactivation of myofibroblasts [2]. This progressive parenchymal fibrosis will disrupt the structure and function of the gas-exchanging region in the lung, and eventually lead to respiratory failure and death [3]. According to the causes of the disease, PF was divided into two categories: one is secondary pulmonary interstitial disease, which has a clear etiology, including cystic fibrosis, pulmonary fibrosis after virus infection, etc.; the other is idiopathic pulmonary interstitial disease, which is not clear in etiology, and the disease with the highest mortality is IPF [1,2,3,4]. In 2015, a guideline indicated that Pirfenidone and Nintedanib are the only two drugs recommended for the treatment of IPF [4,5], and they can alleviate the decline in lung function; however, neither of these drugs actually improves or even stabilizes the disease or the symptoms perceived by the patient [6]. Therefore, we still need to continue to develop safe and effective treatment drugs for PF.

Many cytokines are involved in the development of PF, and TGF-β1 is the central regulator in the progression of PF [7,8]. By regulating its downstream signaling pathways, TGF-β1 plays a vital role in promoting the migration, proliferation and differentiation of fibroblasts, aggravating the injury of epithelial cells and other key processes in PF [7,8,9]. However, due to the pleiotropy of TGF-β1 in normal tissue homeostasis, the complete inhibition of TGF-β1 in PF treatment will have serious side effects [10,11,12]. Therefore, targeting the downstream of the TGF-β1 signaling pathway or its “crosstalk” in the signal transduction pathway is better. JAK-STAT is a downstream signaling pathway regulated by TGF-β1 [7,9], and recent studies have shown that the JAK-STAT pathway is abnormally activated during the process of PF [7]. Javier Milara et al. reported that the JAK2 pathway is involved in the activation of myofibroblasts and lung epithelial cell injury in PF [13]. Research also confirmed that the activation of the JAK-STAT signaling pathway plays a key role in the expression of TGF-β1 downstream genes in hepatic stellate cells, thereby promoting the progression of liver fibrosis [14,15]. Meanwhile, other studies have shown that TGF-β-induced activation of STAT3 signaling could promote the activation of fibroblasts and tissue fibrosis [16,17]. Therefore, targeting the JAK-STAT signaling pathway may be a potential therapeutic method for PF.

At present, the research on disease and drug innovation related to the JAK-STAT signaling pathway mainly focuses on inflammation and tumor diseases [18]. Baricitinib is a JAK1/2 inhibitor with strong anti-inflammatory activity and is clinically used in the treatment of active rheumatoid arthritis [19]. Currently, baricitinib is also being evaluated in clinical trials of COVID-19 infection [20,21,22]. In this study, we established in vivo and in vitro models to analyze the effect of JAK1/2 inhibitor baricitinib in pulmonary fibrosis.

## 2. Results

### 2.1. Baricitinib Attenuates BLM-Induced Pulmonary Fibrosis in Mice

In order to validate the therapeutic effect of baricitinib on pulmonary fibrosis, C57BL/6J mice were given bleomycin (2U) by intratracheally injection and then different doses of baricitinib (10, 20, 40 mg/kg) were given by gavage for 7 days (the seventh day to the fourteenth day), and Nintedanib (100 mg/kg) was used as a positive control (Figure 1A). We observed that the content of hydroxyproline and the area of fibrosis in the treatment group were significantly decreased (Figure 1B,C). In addition, we also used H&E and Masson trichrome staining to evaluate the severity of lung fibrosis in mice. As indicated in Figure 1D, baricitinib treatment could significantly suppress the collagen deposition. Pulmonary function is an important evaluating indicator of pulmonary fibrosis. As shown in Figure 1E–H, baricitinib significantly improved the lung function of mice with bleomycin-induced pulmonary fibrosis, as seen by increased forced vital capacity (FVC) and dynamic compliance (CYDN), and it reduced inspiratory resistance (RI) and expiratory resistance (RE). Based on the above data, we concluded that baricitinib could improve BLM-induced pulmonary fibrosis in mice.

### 2.2. Baricitinib Suppresses TGF-β1-Induced Proliferation and Migration of Lung Fibroblasts

Next, we detected whether baricitinib could affect the proliferation and migration of fibroblasts. The results of MTT assays showed that, without TGF-β1 stimulation, baricitinib had no significant toxic effect on fibroblasts at <10 μM (Figure 2A) but could dose-dependently suppress the proliferation of fibroblasts activated by TGF-β1 (Figure 2B). Furthermore, the wound healing experiment was used to detect the effect of baricitinib on the migration of activated fibroblasts induced by TGF-β1, and the results indicate that baricitinib dose-dependently decreased the migration rate of activated fibroblasts (Figure 3A,B).

### 2.3. Baricitinib Attenuates TGF-β1-Induced Activation of Lung Fibroblasts

We investigated the effect of baricitinib on fibroblast activation. As shown in Figure 4A,B, baricitinib could inhibit TGF-β1-induced protein expression of α-SMA, COL I and FN in a dose-dependent manner. Similarly, baricitinib could reduce the gene expression level of TGF-β1-induced α-SMA, COL I and FN in a dose-dependent manner (Figure 4C,D). The results of the immunofluorescence assay proved that the expression level of α-SMA was decreased in the baricitinib-treatment group (Figure 4E,F). The above data confirmed that baricitinib could inhibit the activation and suppress the ECM production of myofibroblasts.

Previous studies have confirmed that TGF-β1-Smad and non-Smad signaling pathways play important roles in regulating the proliferation and activation of pulmonary fibroblasts [23,24,25,26,27]. Thus, we used Mlg cells and NIH-3T3 cell lines to detect the effect of baricitinib on the TGF-β1 signaling pathway, while the results indicate that there was no significant change in the Smad signaling activity after treatment with baricitinib (Figure 5A,B). Meanwhile, we also detected the levels of proteins in Akt and MAPK pathways, including phosphorylated P38, ERK, JNK and AKT in TGF-β1-treated Mlg cells and NIH-3T3 cells, and the results reveal that baricitinib could restrain the activation of the TGF-β1/non-Smad signaling pathway (Figure 4C,D).

### 2.4. Baricitinib Inhibits the Activation of Lung Fibroblasts *In Vivo*

In order to further evaluate the effect of baricitinib on the activation of lung fibroblasts in vivo, the expression levels of α-SMA, COL I and FN in lung tissues of bleomycin-injured mice were measured by immunohistochemistry, Western blot and qRT-PCR. Immunohistochemical studies indicated that the expression levels of α-SMA, COL I and FN in lung tissues of fibrosis mice were significantly decreased after treatment with baricitinib (Figure 6A). In addition, the Western blotting and qRT-PCR results show that the protein and gene expression levels of α-SMA, COL I and FN in the lung tissues of baricitinib-treated mice were decreased (Figure 6B,C). Accordingly, these results suggested that baricitinib could ameliorate BLM-induced pulmonary fibrosis in mice by inhibiting the activation of fibroblasts in vivo.

### 2.5. Baricitinib Inhibits TGF-β1-Induced Epithelial Injury in Alveolar Epithelial Cells

A current model of IPF suggests that recurrent injury to epithelial cells and ineffective repair may disrupt epithelial–mesenchymal interactions, thus leading to mesenchymal change and aberrant fibroblastic responses, and TGF-β1 plays a central regulatory role in the process. Therefore, we used a cell model in which TGF-β1 stimulated mesenchymal transformation of epithelial cells to evaluate the drug effect [3,28]. In order to investigate whether baricitinib can exert an anti-fibrotic effect via suppressing epithelial injury, we used TGF-β1 to treat lung epithelial cells and co-treated with different doses of baricitinib (300, 600, 900 nM). After 24 h, the morphological changes of A549 cells and the expression of epithelial/mesenchymal phenotype markers were detected. The results show that TGF-β1 could prolong the pseudopodia of A549 cells and produce fibroblast-like morphological changes, while baricitinib could reverse the morphological changes and epithelial–mesenchymal transition in varying degrees (Figure 7A). The qRT-PCR analysis showed that baricitinib could increase the expression of epithelial marker (E-Cadherin) and reduce the expression of mesenchymal markers (Vimentin and N-Cadherin) in TGF-β1-treated MLE12 and A549 cells (Figure 7B,C). Consistent with the qRT-PCR results, the results of Western blot and immunofluorescence analysis showed similar results (Figure 7D–G). In conclusion, baricitinib could inhibit the injury of alveolar epithelial cells induced by TGF-β1.

### 2.6. Baricitinib Alleviates Epithelial Injury *In Vivo*

Based on the in vitro studies, we further studied the effect of baricitinib on epithelial injury in vivo. Immunohistochemical analysis showed that baricitinib treatment leading to the increased expression of E-cadherin and decreased the expression of Vimentin in BLM-injured lung tissues (Figure 8A). Additionally, similar conclusions were drawn in the Western blot and RT-PCR assays (Figure 8B,C). Based on the above results, we concluded that baricitinib could reduce epithelial injury in vivo.

### 2.7. Baricitinib Alleviates Pulmonary Fibrosis by Inhibiting Activation of JAK-STAT Signaling Pathway

Previous studies have confirmed that JAK-STAT signaling pathway plays an important role in proliferative diseases such as pulmonary fibrosis, liver fibrosis and systemic sclerosis [13,16,17,29]. Therefore, we are here to explore whether baricitinib could regulate the activity of the JAK-STAT signaling pathway in the progression of pulmonary fibrosis. Firstly, we analyzed the effect of baricitinib on JAK-STAT signaling pathway in vivo by immunohistochemistry and Western blot. After treatment with baricitinib, the phosphorylation level of JAK1 was decreased in a dose-dependent manner (Figure 9A). Consistent with immunohistochemical results, the phosphorylation level of JAK2 and STAT3 was decreased after treatment with baricitinib (Figure 9B). Next, we evaluated the activation of JAK1/2 and STAT3 in alveolar epithelial cells. Western blotting results indicate that baricitinib attenuated the mesenchymal transformation of epithelial cells through inhibiting the activation of the JAK1/2-STAT3 signaling pathway (Figure 9C,D). Based on the above experimental results, we concluded that baricitinib could alleviate pulmonary fibrosis by inhibiting the activation of JAK1/2-STAT3 signaling pathways.

## 3. Materials and Methods

### 3.1. Materials

The details of reagents used in this study were listed in Table 1.

### 3.2. Cell Culture

The details of cell lines used in this study were listed in Table 2.

### 3.3. Animals

Male C57BL/6 mice (6–8 wk, 20–25 g) were purchased from Academy of Military Medical Sciences of People’s Liberation Army (Beijing, China). Animal testing protocols comply with National Institutes of Health guidelines (NIH Publications No. 85-23, revised 1996). All mice were maintained and handled and experimental procedures are in accordance with the guidelines approved by the Institutional Animal Care and Use Committee (IACUC) of Nankai University (Permission No. SYXK 2019-0001).

### 3.4. Mouse Models of Pulmonary Fibrosis

As mentioned above, we selected 6–8-week-old male C57BL/6J mice and established a bleomycin-induced animal model of pulmonary fibrosis by a single intratracheal instillation of BLM (2 U) dissolved in saline (0.9% NaCl) using a high-pressure syringe as described previously [23]. Additionally, 0.9% NaCl was injected as control. A total of 42 mice were randomly divided into six groups, with seven mice in each group: NaCl group, BLM group, positive control group (BLM+Nintedanib, 100 mg/kg), low-dose Baricitinib (BLM+Baricitinib, 10 mg/kg), medium-dose Baricitinib (BLM+Baricitinib, 20 mg/kg) and high-dose Baricitinib (BLM+Baricitinib, 40 mg/kg). The treatment group was given intragastric administration. Drugs were suspended in double-distilled water, while control and BLM groups were given equal volumes of solvent (double-distilled water). Mice were harvested on the fourteenth day after BLM stimulation, and the left lung was used for pathological staining, and the right lung was used for hydroxyproline and fibrosis marker detection. Finally, the therapeutic effect of baricitinib on BLM-induced pulmonary fibrosis was statistically analyzed.

### 3.5. Hydroxyproline Assay

Hydroxyproline (HYP) was tested as previously described [24]. Additionally, 5 mL ampoule bottles containing the right lung tissues of mice were placed in an oven of 120 °C. After about 16 h (the lung tissues were dried black and could be shaken freely), the ampoule bottles were taken out from the oven to cool the tissue samples to room temperature. Then, 3 mL (6 mol/L) of hydrochloric acid was added to each sample and the cap was tightened. Afterwards, all samples were placed in an oven set at 120 °C for acid hydrolysis for 6 h. The tissue samples were then removed from the oven, brought to room temperature, and their pH adjusted from 6.5 to 7.5 using 6 mol/L NaOH and 6 mol/L HCl. Each sample was then filtered through a 5 m filter, and the volume of each sample was brought to 10 mL with 1× PBS. After preparation, a series of samples with a concentration of 0, 0.625, 1.25, or 2.5 μg/μL were prepared using the concentration gradient dilution method (hydroxyproline stock standard solution 100 μg/μL). Operate according to the instructions, and then determine the OD value at 570 nm wavelength. According to the OD value, make the standard curve and obtain the formula, so as to obtain the hydroxyproline concentration of the measured sample. The total amount of hydroxyproline in the right lung of each mouse was W:W = C × 8 (dilution multiple of measured sample) × 10 (total sample volume).

### 3.6. Histological Examination

After fixation of left lung tissue in 10% neutral formalin for 48 h, samples were dehydrated and paraffin-embedded. Paraffin-embedded tissue blocks were cut into 4 μm thick sections for hematoxylin and eosin (H&E) staining and 5 μm thick for immunohistochemistry and Masson staining. H&E staining: After dehydration, the sections were stained with hematoxylin for 8 min, soaked in water for 10 min, 95% ethanol for 3 min, eosin for 2 min, and finally dehydrated and mounted. Masson Stain: Follow the instructions of the Solarbio kit.

Fibrosis area statistics: Take photos with 3D histact full-automatic scanner, identify the total area of lung tissue with SV software, circle the area of pulmonary fibrosis area, (fibrosis ratio = fibrosis area total pixels/total lung total pixels) and calculate the proportion of pulmonary fibrosis area.

IHC staining densities: First, change the IHC image into a grayscale image in ImageJ software, then convert the 8-bit grayscale image into an OD value. After selecting the parameters to be measured, select the positive signal area (red means selected), and finally get the positive integrated optical density value IOD (integrated optical density).

### 3.7. Pulmonary Function Testing

The mice’s trachea was exposed after general anesthesia, and a tracheal cannula was timed and fixed. According to the manufacturer’s instructions, the mice were moved to a plethysmography chamber for pulmonary function study using an Anires2005 system (Beijing Biolab, Beijing, China).

### 3.8. Cell Viability Analysis

Cell viability was measured using 3-(4,5-dimethylthiazol-2-yl)-2,5-diphenyltetrazolium bromide (MTT) as previously described [24]. Mlg and NIH-3T3 cells were taken out of the incubator and planted in 96-well plates, so that the cell density reached about 30~40%. Then, different concentrations of baricitinib (0 to 160 µM) were added. After that, the 96-well plates were moved to incubate at 37 °C for 24 h according to the experimental requirements. After the incubation time, the cell morphology of each group was observed under the microscope. After that, 20 µL of MTT (5 mg/mL) was added and incubated for 4 h. Finally, 200 µL DMSO was added to each well and the absorbance was read using a microplate reader at 570 nm.

### 3.9. Wound-Healing Assay

Then, wound-healing assays were performed according to previous methods [24]. NIH-3T3 cells and Mlg cells were seeded in a 24-well plate containing 10% FBS medium. When the cell confluence is above 90%, draw a straight line perpendicular to the bottom line in each hole with the 200 µL pipette tip. Then, sterilized 1 × PBS was used to wash three times (the last time residual liquid was aspirated thoroughly), and then 0.1% FBS medium was added. At the same time, different concentrations of baricitinib and TGF-β1 were added to each well. At 0, 6, 12 and 24 h, the wound was photographed with a microscope and recorded. The cell migration distance was statistically analyzed by Image J, and the cell migration rate was finally obtained.

### 3.10. Quantitative Real-Time PCR (qRT-PCR)

According to a previously established protocol [24], Quantitative real-time PCR was used to measure the level of gene expression using the LightCycler 96. TRIzol Reagent was used to extract RNA from cultivated cells and lung tissues in accordance with the manufacturer’s instructions. FastKing gDNA Dispelling RT SuperMix was used to reverse-transcribe the RNA, which was then ready for qRT-PCR. The LightCycler 96 Real-Time PCR System was used for RT-PCR detection. All primers were made by Tsingke Biotechnology. Target gene relative mRNA levels were normalized to GAPDH or β-actin mRNA levels. The target gene primer pairs used were listed in Table 3.

### 3.11. Western Blot Analysis

The changes of target proteins in both cells and tissues were measured by Western blot. All proteins were extracted from cells or tissues. SDS-PAGE gel electrophoresis was used to separate the total protein samples, and the results were then transferred to PVDF membrane. Following blocking, the following primary antibodies were used to investigate the immunoblots: α-SMA (Santa Cruz, Beijing, China), Collagen I (Affinity, Shanghai, China), Fibronectin (Affinity, Shanghai, China), β-tubulin (Proteintech, Wuhan, China), GAPDH (Cell Signaling Technology, Boston, MA, USA), Phospho-P38 (Affinity, Shanghai, China), P38 (Affinity, Shanghai, China), Phospho-JNK (Affinity, Shanghai, China), JNK (Affinity, Shanghai, China), Phospho-ERK (Affinity, Shanghai, China), ERK (Affinity, Shanghai, China), Phospho-AKT (Affinity, Shanghai, China), AKT (Affinity, Shanghai, China), E-cadherin (Cell Signaling Technology, Boston, MA, USA), N-cadherin (Cell Signaling Technology, Boston, MA, USA), Vimentin (Cell Signaling Technology, Boston, MA, USA), Phospho-JAK1 (Cell Signaling Technology, Boston, USA), JAK1 (Cell Signaling Technology, Boston, USA), Phospho-JAK2 (Cell Signaling Technology, Boston, MA, USA), JAK2 (Cell Signaling Technology, Boston, MA, USA), Phospho-STAT3 (Cell Signaling Technology, Boston, MA, USA), STAT3 (Cell Signaling Technology, Boston, MA, USA). The membranes were cut horizontally and examined with chemiluminescence reagent after being treated with HRP-conjugated secondary antibodies for 2 h at room temperature (Affinity, Shanghai, China). MW of proteins above were marked in Figure legends.

### 3.12. Immunofluorescence

The cells were cultured on 24-well plates with sliders in a medium containing 10% FBS. When the cell density reached 50%, the culture medium was changed to the medium containing 0.1% FBS. Different concentrations of baricitinib and TGF-β1 were added and treated for 24 h. 4% pareformaldehyde fixative solution (PFA) (Solarbio, Beijing, China) was added for fixation (500 µL, 15 min), and 0.2% Triton X-100 (500 µL, 20 min) was added for penetration after washing. After 30 min of 5% BSA occlusion (in order to reduce nonspecific binding), primary antibody was incubated and refrigerated overnight at 4 °C. The next day, they were incubated at room temperature for 30 min, and then washed with 1 × PBST 3 times for 5 min each time. Then, the TRITC-conjugated or FITC-conjugated secondary antibody was used to incubate at room temperature for 2 h (the process needed to be kept away from light to avoid the failure of fluorescent secondary antibody), and then washed with 1 × PBST for 3 times, 5 min each time. Finally, a small amount of DAPI (nuclei stained blue) was added to the slide to seal the slide. Images were taken by laser scanning confocal microscopy (Leica SP8, Wetzlar, Germany).

### 3.13. Immunohistochemistry Staining

Lung tissue that had been paraffin-embedded was dewaxed with xylene before sections were put in a microwave with antigen-fixing solution (0.01 M citrate buffer) and taken out after 20 min. The primary antibody was incubated at 4 °C overnight after being cooled to room temperature, being washed three times for three minutes with PBS, and being blocked with an immunohistochemistry kit. The primary antibodies were as follows: mouse anti-α-SMA antibody (1:200 dilution), rabbit anti-collagen I antibody (1:200 dilution), mouse anti-Fibronectin antibody (1:200 dilution), mouse anti-E-Cadherin (1:200 dilution) and mouse anti-Vimentin (1:200 dilution). After washing with PBST three times, the sections were incubated with the secondary antibody at room temperature for 1 h. Next, DAB staining was performed according to the kit instructions. Double-distilled H_2_O was used to stop staining and PBS was used to rinse the tissue sections three times for five minutes. Finally, hematoxylin staining was used to show the location of the nucleus, and then the histological changes and target protein expression in the mouse lung tissue could be detected under a light microscope.

### 3.14. Statistical Analysis

All cell culture experiments obtained similar results from three independent replicates. Data are presented as mean ± SD using Prism version 7.0 software. Statistical differences between two groups (administration group and control group) were calculated using Student’s *t*-test, and significant differences between multiple groups were calculated using one-way analysis of variance with subsequent Bonferroni correction. *p* values less than 0.05 were considered significant.

## 4. Discussion

Pulmonary fibrosis is a progressive pulmonary disease with high mortality and limited therapy [4,12]. Previous studies have reported the pro-fibrotic activity of JAK2 and STAT3 in pulmonary fibrosis [13,16,17], but the involvement of JAK1 has not been well studied. Recently, Zhang et al. reported that JAK2 inhibitors have a weakened effect during long-term anti-pulmonary fibrosis treatment, but simultaneous inhibition of JAK1 and JAK2 can effectively improve the anti-fibrotic effect. This study examined the role of JAK1/2 inhibitor baricitinib in pulmonary fibrosis. Our results show that JAK1/2 and STAT3 were activated in animal and cell models of pulmonary fibrosis and that baricitinib has therapeutic effects on pulmonary fibrosis by selectively inhibiting JAK1/2, including reducing the collagen deposition and improving pulmonary function in BLM-injured mice. Baricitinib also could inhibit the proliferation and activation of myofibroblasts, and reduce the injury of epithelial cells.

Studies have shown that BLM could induce pulmonary fibrosis in mice, and the pathological manifestations are similar to clinical manifestations of IPF patients [30], and the model is available and repeatable, which meets the standards of animal model [31]. Mice were strain dependent on BLM-induced pulmonary fibrosis, and C57BL/6J mice were more sensitive to BLM-induced pulmonary fibrosis [32,33]. Therefore, it is feasible to use BLM model of C57BL/6J mice to study the effect of baricitinib on pulmonary fibrosis in vivo.

TGF-β1 is a well-studied profibrotic growth factor and plays a vital role in the regulation of pulmonary fibrosis by driving the activation of pulmonary fibroblasts and promoting the transformation of epithelial cells [7,34]. In this study, we used fibroblast and epithelial cell models stimulated by TGF-β1 to investigate the effect of baricitinib on pulmonary fibrosis in vitro. After being stimulated by TGF-β1, the expression levels of α-SMA, COL I and FN were increased in fibroblasts, while the expression levels were inhibited after treatment with baricitinib. The expression of epithelial markers was suppressed, and the expression of mesenchymal markers was elevated after stimulation with TGF-β1, while baricitinib could reverse this phenomenon.

In this study, our results suggest that baricitinib may play an anti-fibrotic role through inhibiting the JAK-STAT signaling pathway. The literature suggests that JAK2 is phosphorylated upon stimulation of various cytokines or growth factors, including TGF-β1, leading to the translocation of STAT3 to the nucleus, where it activates fibrosis-related genes [13,16,17,29]. Recently, there is evidence that in fibroblasts of patients with IPF, TGF-β1 activates STAT3 signaling in a Smad2/3-dependent way, which is independent of JAK2; Liu-Ya Tang et al. found that JAK1, as a TGF βRI-interacting protein, can mediate TGF-β1, which induces early phosphorylation of STAT3 in a manner independent of Smad, thus promoting liver fibrosis [35]. Therefore, it is speculated that as a JAK1/2 inhibitor, baricitinib may not respond to the Smad pathway, and then we verified that baricitinib on the TGF-β/Smad pathway was ineffective by Western blot, but the Western blot results show that it could play an anti-fibrosis role in fibroblasts through the non-Smad pathway.

A recent study showed that severe COVID-19 patients with primary thrombocythemia and IPF were significantly improved by the treatment of JAK1/2 inhibitor Ruxolitinib [36,37]. At present, baricitinib is in the clinical trial of COVID-19. The NCT04401579 trial showed that the median days of mechanical ventilation and ECMO in patients with COVID-19 infection after the combined use of baricitinib and remdesivir were significantly lower than those of remdesivir alone. Clinical trials have also shown that patients’ respiratory function has improved after treatment with baricitinib [38]. Therefore, it is reasonable to infer that baricitinib, a JAK1/2 inhibitor, may also play an anti-fibrosis role in IPF patients infected with COVID-19, and baricitinib can be developed as a drug candidate to effectively prevent pulmonary fibrosis after COVID-19 infection.

In summary, our results show that JAK1/2 inhibitor baricitinib had the efficacy on restraining pulmonary fibrosis progression by suppressing TGF-β1-induced fibroblasts activation and epithelial injury (Figure 10).

## Figures and Tables

**Figure 1 molecules-28-02195-f001:**
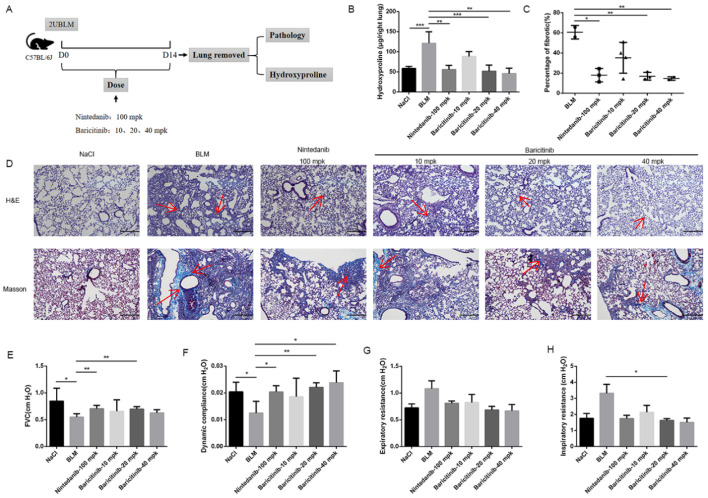
Baricitinib ameliorates BLM-induced pulmonary fibrosis in mice. (**A**) Dosing regimen in BLM-induced pulmonary fibrosis model. (**B**) HYP contents of lung tissues in mice. (**C**) Statistics of lung fibrosis area among groups. (**D**) Lung tissue sections were stained with hematoxylin-eosin (HE) and Masson Trichrome staining. Red arrows indicated collagen deposition. (**E**) Forced vital capacity (FVC) of mice. (**F**) Dynamic compliance of mice. (**G**) Expiratory resistance of mice. (**H**) Inspiratory resistance of mice. Scale bar = 50 μm. Data were noted as the means ± SD, *n* = 6. * *p* < 0.05, ** *p* < 0.01, *** *p* < 0.001.

**Figure 2 molecules-28-02195-f002:**
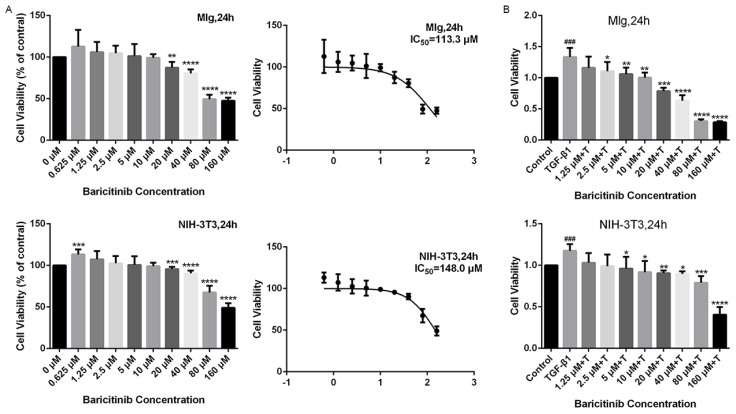
Baricitinib suppresses TGF-β1-induced proliferation of fibroblasts. (**A**) MTT assays of Mlg cells and NIH-3T3 cells. Cells were exposed to the indicated doses of baricitinib (0 to 160 μM) for 24 h, IC50 = 113.3 and 148.0 μM, (*n* = 5 per group). (**B**) MTT assays were performed to test the effect of baricitinib on the proliferation of TGF-β1-stimulated Mlg cells and NIH-3T3 cells. Mlg cells and NIH-3T3 cells were treated with baricitinib (0 to 160 μM) and TGF-β1 (5 ng/mL) for 24 h, (*n* = 5 per group). Data were presented as the means ± SD, *n* = 5. * represent the difference between TGF-β1-treated group and baricitinib treatment group, # indicated the difference between TGF-β1 treated group versus the control group. * *p* < 0.05, ** *p* < 0.01, *** *p* < 0.001, **** *p* < 0.0001, ### *p* < 0.001.

**Figure 3 molecules-28-02195-f003:**
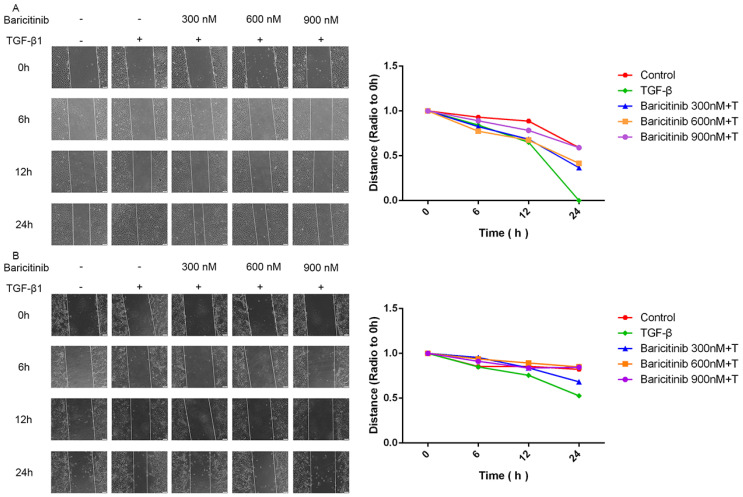
Baricitinib suppresses TGF-β1-induced migration of fibroblasts. (**A**,**B**) Wound healing assays of Mlg and NIH-3T3 cells co-cultured with TGF-β1 (5 ng/mL) and baricitinib (300, 600, 900 nM). The wound closure was photographed at 0, 6, 12 and 24 h post-scratching. Data were presented as the means ± SD, *n* = 3.

**Figure 4 molecules-28-02195-f004:**
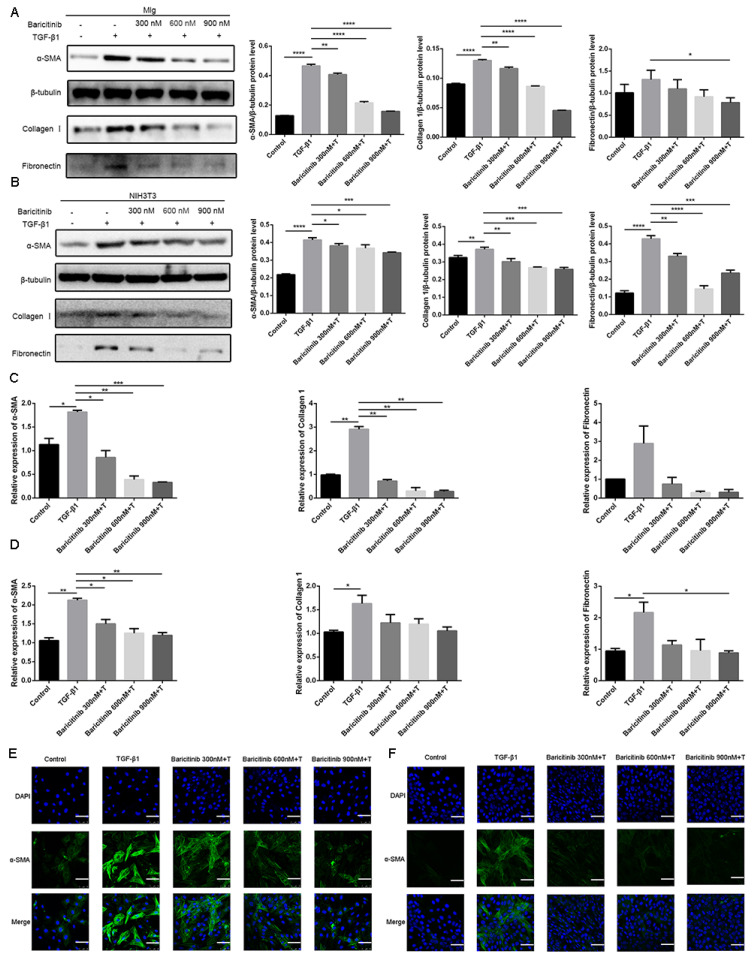
Baricitinib attenuates TGF-β1-induced fibroblasts activation. (**A**,**B**) Mlg cells and NIH-3T3 cells were treated with TGF-β1 (5 ng/mL) and baricitinib (300, 600, 900 nM) for 24 h. Mlg (**A**) and NIH-3T3 (**B**) cells were extracted for Western blot analysis of α-SMA, collagen I and fibronectin. (**C**,**D**) Mlg cells and NIH-3T3 cells were treated with TGF-β1 (5 ng/mL) and baricitinib (300, 600, 900 nM) for 24 h. α-SMA (43 kD), β-Tubulin (52 kD) collagen I (220 kD) and fibronectin (260 kD) were analyzed by real-time PCR in Mlg (**C**) and NIH-3T3 (**D**) cells. (**E**,**F**) Immunofluorescence staining of α-SMA was performed on Mlg (**E**) and NIH-3T3 (**F**) cells treated with/without TGF-β1 (5 ng/mL) and/or baricitinib (300, 600, 900 nM) for 24 h. Scale bar = 60 μm, Data were presented as the means ± SD, *n* = 3. * *p* < 0.05, ** *p* < 0.01, *** *p* < 0.001, **** *p* < 0.0001.

**Figure 5 molecules-28-02195-f005:**
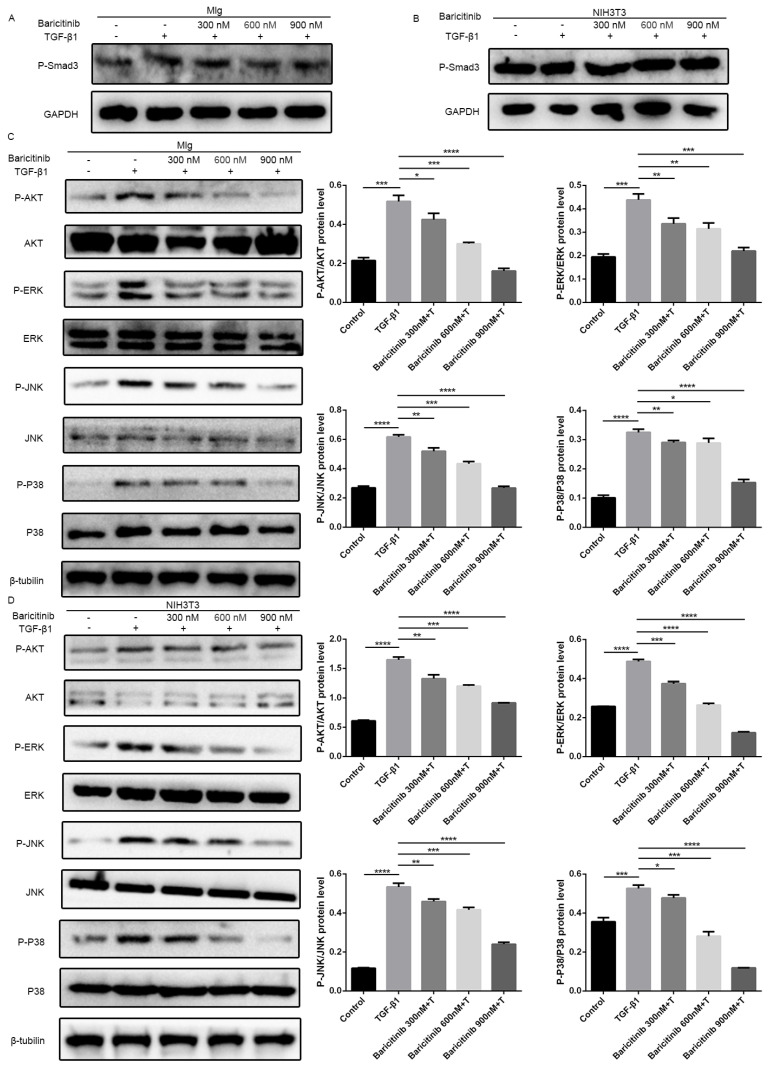
Baricitinib inhibits TGF-β1-induced activation of non-Smad signaling pathway in fibroblasts. (**A**,**B**) Mlg (**A**) and NIH-3T3 (**B**) cells were treated with TGF-β1 (5 ng/mL) and baricitinib (300, 600, 900 nM) for 30 min. P-Smad3 (58 kD) were assessed using Western blot. GAPDH (37 kD) was used as the internal control. (**C**,**D**) The phosphorylation levels of P-38 (42 kD), JNK (54 kD), ERK (43 kD) and AKT (56 kD) were analyzed by Western blot in Mlg (**C**) and NIH-3T3 (**D**) cells treated with TGF-β1 (5 ng/mL) and baricitinib (300, 600, 900 nM) for 1 h. β-tubulin was used as a loading control in grayscale analysis. Data were noted as the means ± SD, *n* = 3. * *p* < 0.05, ** *p* < 0.01, *** *p* < 0.001, **** *p* < 0.0001.

**Figure 6 molecules-28-02195-f006:**
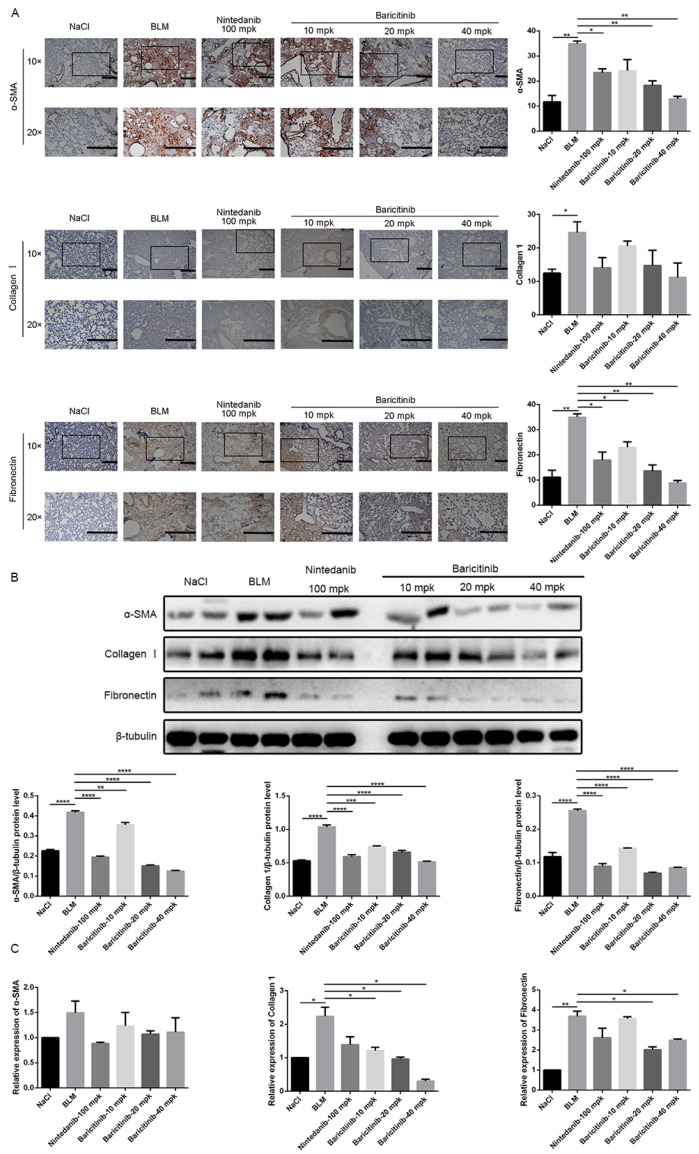
Baricitinib inhibits activation of myofibroblasts in vivo. (**A**) Immunohistochemical staining analysis of α-SMA and collagen I and fibronectin in the lung tissues. Scale bar = 50 μm. (**B**) Protein levels of α-SMA (43 kD), collagen I (220 kD) and fibronectin (260 kD) were verified by Western blot in lung tissues. β-tubulin (52 kD) was used as an internal reference in densitometric analysis. (**C**) RT-PCR was performed to detect mRNA levels of α-SMA and collagen I and fibronectin. Data were presented as the means ± SD, *n* = 3. * *p* < 0.05, ** *p* < 0.01, *** *p* < 0.01, **** *p* < 0.0001.

**Figure 7 molecules-28-02195-f007:**
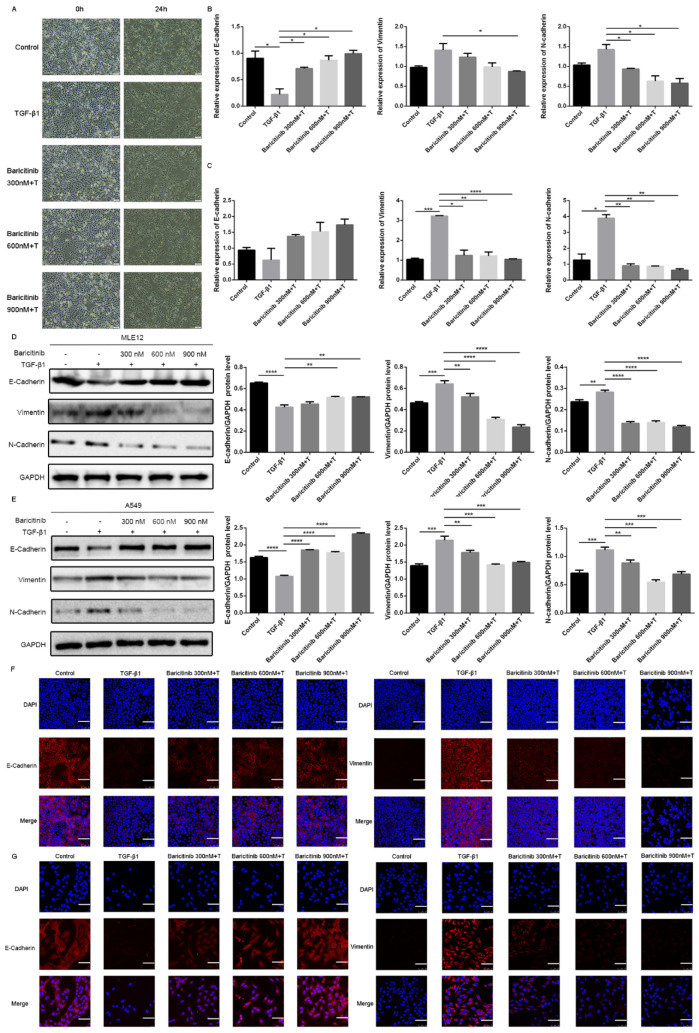
Baricitinib inhibits TGF-β1-induced epithelial injury in alveolar epithelial cells. (**A**) Morphological changes of A549 cells. (**B**,**C**) MLE12 and A549 cells were exposed to Baricitinib for 30 min (300, 600, 900 nM) and treated with TGF-β1 (5 ng/mL) for 24 h. (**B**) mRNA levels E-cadherin, N-cadherin and Vimentin were tested by RT-PCR in MLE12 (**B**) and A549 (**C**) cells. (**D**,**E**) MLE12 and A549 cells were treated with TGF-β1 (5 ng/mL) and baricitinib (300, 600, 900 nM) for 24 h. Protein expression levels of E-cadherin (120 kD), Vimentin (53 kD), and N-cadherin (100 kD) were assessed by Western blot in A549 (**C**) and MLE12 (**D**) cells. GAPDH (37 kD) was used as a loading control. (**F**,**G**) MLE12 (**F**) and A549 (**G**) cells treated with TGF-β1 (5 ng/mL) and baricitinib (300, 600, 900 nM) for 24 h were immune-stained with E-cadherin and Vimentin. Scale bar = 60 μm. Data were presented as the means ± SD, *n* = 3. * *p* < 0.05, ** *p* < 0.01, *** *p* < 0.001, **** *p* < 0.0001.

**Figure 8 molecules-28-02195-f008:**
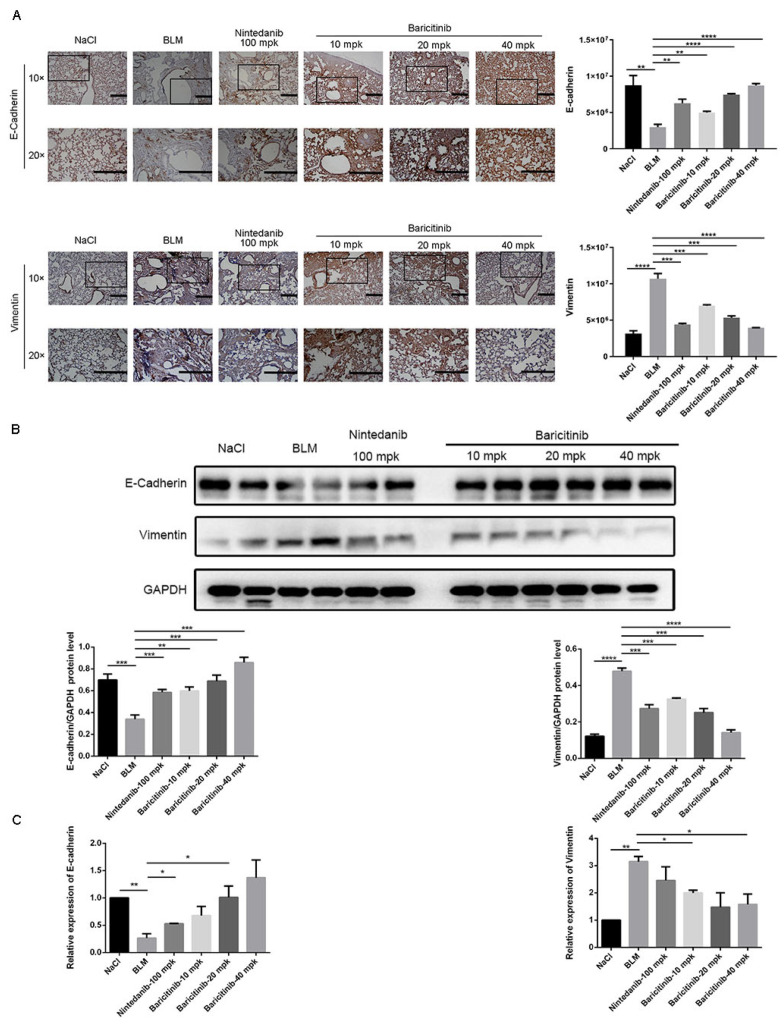
Baricitinib alleviates epithelial injury in vivo. (**A**) IHC staining of E-cadherin and Vimentin in the lung tissues. (**B**) Protein levels of E-cadherin (120 kD), GAPDH (37 kD) and Vimentin (53 kD) in the lung tissues. (**C**) mRNA expression of E-cadherin and Vimentin in the lung tissues. Scale bar = 50 μm. Data were noted as the means ± SD, *n* = 3. * *p* < 0.05; ** *p* < 0.01; *** *p* < 0.001; **** *p* < 0.0001.

**Figure 9 molecules-28-02195-f009:**
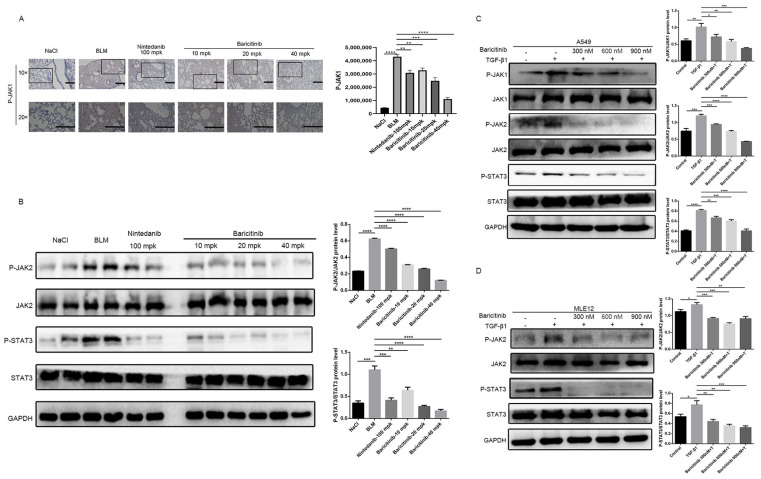
Baricitinib alleviates pulmonary fibrosis by inhibiting activation of JAK-STAT signaling pathway. (**A**) Immunohistochemical staining analysis of P-JAK1 in the lung tissues. Scale bar = 50 μm. (**B**) The phosphorylation levels of JAK2 and STAT3 were analyzed by Western blotting the lung tissues. (**C**) The phosphorylation levels of JAK1, JAK2 and STAT3 were analyzed by Western blot in A549 cells treated with TGF-β1 (5 ng/mL) and baricitinib (300, 600, 900 nM) for 1 h. GAPDH (37 kD) was used as a loading control in grayscale analysis. (**D**) The phosphorylation levels of JAK2 (125 kD) and STAT3 (88 kD) were analyzed by Western blot in MLE12 cells treated with TGF-β1 (5 ng/mL) and baricitinib (300, 600, 900 nM) for 2 h. GAPDH was used as a loading control in grayscale analysis. Data were noted as the means ± SD, *n* = 3. * *p* < 0.05; ** *p* < 0.01; *** *p* < 0.001; **** *p* < 0.0001.

**Figure 10 molecules-28-02195-f010:**
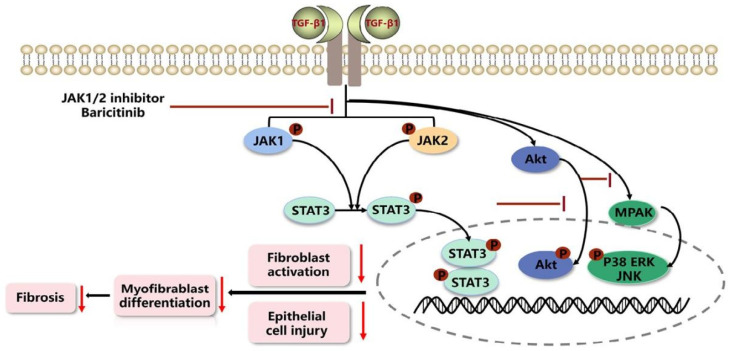
Mechanism for the anti-pulmonary fibrosis effect of Baricitinib.

**Table 1 molecules-28-02195-t001:** The details of reagents.

Reagents	Manufacturer
Recombinant Human TGF-β1	Peprotech (Cranbury, NJ, USA)
Collagen Ⅰ, Fibronectin, P-JAK1/JAK1, P-JAK2/JAK2, P-STAT3/STAT3, P-AKT/AKT, P-ERK/ERK, P-JNK/JNK, P-P38/P38, GAPDH, E-cadherin, Vimentin, N-cadherin antibodies	Cell Signaling Technology (Boston, MA, USA)
β-tubulin antibody	Proteintech (Wuhan, China)
α-SMA antibody	Santa Cruz Biotechnology (Beijing, China)
Goat pAb to Rb IgG (HRP) and Rb pAb to Ms IgG	ImmunoWay (Beijing, China)
TRIzol reagent	Ambion Life Technology (Beijing, China)
DEPC Treated H_2_O and RNAse Away H_2_O	Life Technologies (Shanghai, China)
Fastking gDNA Dispelling RT SuperMix	TIANGEN (Beijing, China)
UNICON^®^ Qpcr SYBR Green Master Mix and Liposomal transfection reagent	YEVSEN (Shanghai, China)
BLM	Nippon Kayaku (Tokyo, Japan)
Masson’s Trichrome Stain Kit	Solarbio (Beijing, China)

**Table 2 molecules-28-02195-t002:** The details of cell lines.

Cell lines	Culture Conditions
Mouse pulmonary epithelial cells (MLE12, ATCC)	DMEM/F-12 (1:1) basic (1×) (Gibco) with 10% fetal bovine serum (FBS, ExCell Bio, Shanghai, China) and 1% antibiotics (gibco)
Human pulmonary epithelial cells (A549, ATCC)	RPMI-1640 (KeyGEN BioTECH, Beijing, China) with 10% FBS
Mouse lung fibroblast cells (Mlg, ATCC)	DMEM (KeyGEN BioTECH, Nanjing, China) with 10% FBS
Mouse embryonic fibroblast cells (NIH-3T3, ATCC)	DMEM (KeyGEN BioTECH, Nanjing, China) with 10% FBS

All cells were incubated with 5% CO_2_ at 37 °C.

**Table 3 molecules-28-02195-t003:** Primers sequences for real-time PCR.

**For Mouse**		
Gene	Forward Primer Sequence (5′-3′)	Reverse Primer Sequence (5′-3′)
GAPDH	TGGA TTTGGACGCA TTGGTC	TTTGCACTGGTACGTGTTGAT
α-SMA	TGGGTGAACTCCATCGCTGTA	GTCGAATGCAACAAGGAAGCC
Collagen I	AAGCCGGAGGACAACCTTTTA	GCGAAGAGAATGACCAGATCC
Fibronectin	GTGCCCGGAATACGCATGTA	CTGGTGGACGGGATCATCCT
E-cadherin	CAGCCTTCTTTTCGGAAGACT	GGTAGACAGCTCCCTATGACTG
Vimentin	GCTGCGAGAGAAATTGCAGGA	CCACTTTCCGTTCAAGGTCAAG
N-cadherin	CTCCAACGGGCATCTTCATTAT	CAAGTGAAACCGGGCTATCAG
**For Human**		
Gene	Forward Primer Sequence (5′-3′)	Reverse Primer Sequence (5′-3′)
β-actin	AGGCCAACCGTGAAAAGATG	AGAGCATAGCCCTCGTAGATGG
E-cadherin	A TTTTTCCCTCGACACCCGAT	TCCCAGGCGTAGACCAAGA
Vimentin	AGTCCACTGAGTACCGGAGAC	CA TTTCACGCA TCTGGCGTTC
N-cadherin	TTTGATGGAGGTCTCCTAACACC	ACGTTTAACACGTTGGAAATGTG

## Data Availability

The original contributions presented in the study are included in the article. Further inquiries can be directed to the corresponding author.

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
