# Peer review of "Baricitinib Attenuates Bleomycin-Induced Pulmonary Fibrosis in Mice by Inhibiting TGF-β1 Signaling Pathway"

_molecules, 2023, doi:10.3390/molecules28052195_

Round 1

Reviewer 1 Report

In this manuscript, the authors demonstrate the effect and mechanism of baricitinib on pulmonary fibrosis in vivo and in vitro. Furthermore, the authors suggest that this drug can effectively attenuate bleomycin-induced pulmonary fibrosis in vivo and attenuate TGF-b1-induced fibroblast activation and epithelial cell injury in vitro by inhibiting TGF-b1/JAK/STAT signaling pathways.

The work is very interesting, and the experimental work is satisfactorily addressed. In general, the conclusion is based on several results in this manuscript.

However, I have a few questions and suggestions:

-          In figure 1, panel D, it will be helpful to use some arrows to indicate collagen deposition.

-          Please indicate if in figure 2, panel B, the authors indicate proliferation or cell viability; both events are very different.

-          Please explain how you selected the baricitinib concentration used in migration assays (figure 3).

-          In figure 7, panel A, the images are very small to see any morphological changes described by the authors.

Author Response

Response to Reviewer 1 Comments

Point 1: In figure 1, panel D, it will be helpful to use some arrows to indicate collagen deposition.

Response 1:We thank the reviewers comment, which is helpful for us to improve the article. We have added some arrows in figure 1panel Dto indicate collagen deposition.

Point 2: Please indicate if in figure 2, panel B, the authors indicate proliferation or cell viability; both events are very different.

Response 2: We thank the reviewers comment. In figure 2, panel B, we detected whether baricitinib could affect the proliferation of fibroblasts. MTT assays were performed to test the effect of baricitinib on the proliferation of TGF-β1-stimulated Mlg cells and NIH-3T3 cells. Mlg cells and NIH-3T3 cells were treated with baricitinib (0 to 160 μM) and TGF-β1 (5 ng/ml) for 24 h. The results of MTT assays showed that baricitinib could dose-dependently suppress the proliferation of fibroblasts activated by TGF-β1 (Fig2 B).

Point 3:Please explain how you selected the baricitinib concentration used in migration assays (figure 3).

Response 3:We thank the reviewers comment. According to the article published by Yuji Joyo et al. in Immunol Res, entitled The Janus kinase inhibitor (baritinib) suppresses the rheumatoid arthritis active marker gliostatin/thymidine photosphyrylase in human fibreblast-like synoviocytes[1], baritinib can inhibit the activation of JAK/STAT signal in the IFN-γtreated human fibroblast-like synovial cells at the concentration of 300nM, and the inhibitory effect continues to increase as the concentration increases, when the concentration reaches 1 μM, the inhibition is the strongest. Therefore, we plan to select 300 nM, 600 nM and 900 nM for follow-up research. As shown in Figures 2A and 2B, baricitinib had no obvious toxic effect on normal (NIH-3T3 and Mlg) cells and the IC50 value of baricitinib was 113.3 μM and 148.0 μM. It can be seen that the above three concentrations it has no effect on the cell viability, which belongs to the safe concentration range. Therefore, we finally chose 300nM, 600nM and 900nM concentrations for migration assays and subsequent study.

Reference:

[1] Joyo Y, Kawaguchi Y, Yonezu H, Senda H, Yasuma S, Shiraga H, Nozaki M, Aoyama M, Asai K, Murakami H, Waguri-Nagaya Y. The Janus kinase inhibitor (baricitinib) suppresses the rheumatoid arthritis active marker gliostatin/thymidine phosphorylase in human fibroblast-like synoviocytes. Immunol Res. 2022 Apr;70(2):208-215. doi: 10.1007/s12026-022-09261-4. Epub 2022 Jan 10. PMID: 35014010; PMCID: PMC8917024.

Point 4:In figure 7, panel A, the images are very small to see any morphological changes described by the authors.

Response 4:We thank the reviewers comment. We have added magnified images about the morphological changes of A549 cells in Fig S1 to have a better reading experience. At the same time, we also marked it in the original text.

Reviewer 2 Report

Review comments

In this research article Gu et. al. reported the antifibrotic effect of Baricitinib in idiopathic pulmonary fibrosis (IPF). With in vitro cell culture models and in vivo mice models they showed that Baricitinib ameliorates the bleomycin induced pulmonary fibrosis through inhibition of the JAK-STAT pathway. Although the drug is used for the treatment of rheumatoid arthritis, the authors demonstrated its promises to treat IPF. However, I have the following observations after reading the manuscript:

Major observations

1.     From figure 2A&B it is not clear how the authors concluded that Baricitinib suppressed the growth of the lung fibroblasts induced by TGF-β1. Because in figure 2A, they maintained the cells in different conc. of Baricitinib for 24 hrs. But in figure 2B, they maintained the cells in TGF-β1 for 24 hrs and then treated them with the same concentrations of Baricitinib for only 30 minutes which is not comparable with the results of 2A. Furthermore, the results obtained in 2B is not very different from 2A if they are expressed in % viability as 2A. Don’t know why they expressed the cell viability in % in 2A and in a different way in 2B.

2.     It is not clear how they determined the doses of Baricitinib for wound healing assay and western blot in figure 3 & 4 and subsequent experiments.

3.     For figure 7, it is also unclear why the authors switched the cell culture model from Mlg and NIH-3T3 to A549 and MLE12 cells.

4.     In the JAK-STAT pathway, the activated STATs act as transcription factors which usually also activate the transcription of inhibitors of this pathway such as the SOCS or PIAS as a mechanism of feedback control. Did the authors test the effect of Baricitinib on the expression of those in either mRNA or protein levels?

5.     The loading control GAPDH seems the same in both figures 8b and 9b. Please double check these. Also do the authors have better western blot images for p-JAK1, p-JAK2 and p-STAT3?

6.     In figure 10, the authors depicted the mechanism of action of Baricitinib where they showed that the drug along with inactivation of JAK1/2-STAT3 pathway also blocks Akt and MAPK pathways. But there is no data shown in the manuscript in support of that.

7.     In lines 122-124, please mention clearly the equation and elaborate the method of calculation of the IHC staining densities.

Minor observations:

1.     In line 103, please indicate the concentration of the HCl used.

2.     In line 254, figure legend of Figure 3, there is no panel called C or D. Please correct accordingly.

3.     There are some typos throughout the text. For example- in line 26 it should be attenuate, in line 51 (a vital role not an), in line 172 it should be horizontally, in line 307 (260 kD) etc.

Overall, this is a nice piece of work, the authors did a good job, and the manuscript is well written.

Author Response

Response to Reviewer 2 Comments

Major observations:

Point 1: From figure 2A&B it is not clear how the authors concluded that Baricitinib suppressed the growth of the lung fibroblasts induced by TGF-β1. Because in figure 2A, they maintained the cells in different conc. of Baricitinib for 24 hrs. But in figure 2B, they maintained the cells in TGF-β1 for 24 hrs and then treated them with the same concentrations of Baricitinib for only 30 minutes which is not comparable with the results of 2A. Furthermore, the results obtained in 2B is not very different from 2A if they are expressed in % viability as 2A. Don’t know why they expressed the cell viability in % in 2A and in a different way in 2B.

Response 1:We thank the reviewers comment. Just as shown in figure 2A&B we detected whether baricitinib could affect the proliferation of fibroblasts. Firstly, Mlg cells and NIH-3T3 cells were exposed to the indicated doses of baricitinib (0 to 160 μM) for 24 h to determine whether baricitinib had toxic effect on normal cells. The results showed that baricitinib had no significant toxic effect on fibroblasts at < 10 μM and the IC50 value of baricitinib was 113.3 μM and 148.0 μM (Fig2 A). Then we used MTT assays to test the effect of baricitinib on the proliferation of TGF-β1-stimulated Mlg cells and NIH-3T3 cells. Mlg cells and NIH-3T3 cells were treated with baricitinib (0 to 160 μM) and TGF-β1 (5 ng/ml) for 24 h. The results of MTT assays showed baricitinib could dose-dependently suppress the proliferation of fibroblasts activated by TGF-β1 (Fig2 B).

The description of Mlg cells and NIH-3T3 cells were treated with baricitinib (0 to 160 μM) for 30 mins, followed with TGF-β1 (5 ng/ml) at 24 h in figure legend has been corrected,  it should be Mlg cells and NIH-3T3 cells were treated with baricitinib (0 to 160 μM) and TGF-β1 (5 ng/ml) for 24 h.

As shown in figure 2A&B, In Fig 2A we used Mlg and NIH-3T3 cells to detect whether baricitinib had toxic effect on normal fibroblasts, while in Figure 2B to detect the effect of baretinib on TGF-β1-induced fibroblast proliferation with activated fibroblasts. The results showed that baretinib can effectively inhibit TGF-β1-induced fibroblasts proliferation within the non-toxic concentration range. So we expressed the cell viability in % in 2A and in a different way in 2B.

Point 2: It is not clear how they determined the doses of Baricitinib for wound healing assay and western blot in figure 3 & 4 and subsequent experiments.

Response 2: We thank the reviewers comment. According to the article published by Yuji Joyo et al. in Immunol Res, entitled The Janus kinase inhibitor (baritinib) suppresses the rheumatoid arthritis active marker gliostatin/thymidine photosphyrylase in human fibreblast-like synoviocytes[1], baritinib can inhibit the activation of JAK/STAT signal in the IFN-γtreated human fibroblast-like synovial cells at the concentration of 300nm, and the inhibitory effect continues to increase as the concentration increases, when the concentration reaches 1 μM, the inhibition is the strongest. Therefore, we plan to select 300 nM, 600 nM and 900 nM for follow-up research. As shown in Figures 2A and 2B, baricitinib had no obvious toxic effect on normal (NIH-3T3 and Mlg) cells and the IC50 value of baricitinib was 113.3 μM and 148.0 μM. It can be seen that the above three concentrations it has no effect on the cell viability, which belongs to the safe concentration range. Therefore, we finally chose 300nM, 600nM and 900nM concentrations for migration assays and subsequent study.

Reference:

  • Joyo Y, Kawaguchi Y, Yonezu H, Senda H, Yasuma S, Shiraga H, Nozaki M, Aoyama M, Asai K, Murakami H, Waguri-Nagaya Y. The Janus kinase inhibitor (baricitinib) suppresses the rheumatoid arthritis active marker gliostatin/thymidine phosphorylase in human fibroblast-like synoviocytes. Immunol Res. 2022 Apr;70(2):208-215. doi: 10.1007/s12026-022-09261-4. Epub 2022 Jan 10. PMID: 35014010; PMCID: PMC8917024.

Point 3:For figure 7, it is also unclear why the authors switched the cell culture model from Mlg and NIH-3T3 to A549 and MLE12 cells.

Response 3:We thank the reviewers comment. As we know Mlg and NIH-3T3 are lung fibroblasts while A549 and MLE12 are epithelial cells. A current model of IPF suggests that recurrent injury to epithelial cells and ineffective repair may disrupt epithelial-mesenchymal interactions, thus leads to mesenchymal change and aberrant fibroblastic responses, and TGF-β1 plays a central regulatory role in the process. Therefore we used cell model in which TGF-β1 stimulated mesenchymal transformation of epithelial cells to evaluate the drug effect[1,2]. In order to investigate whether baricitinib can exert anti-fibrotic effect via suppressing epithelial injury, we switched the cell culture model from lung fibroblasts (Mlg and NIH-3T3) to lung epithelial cells (A549 and MLE12).

Reference:

[1]Hardie WD, Hagood JS, Dave V, Perl AK, Whitsett JA, Korfhagen TR, Glasser S. Signaling pathways in the epithelial origins of pulmonary fibrosis. Cell Cycle. 2010 Jul 15;9(14):2769-76. doi: 10.4161/cc.9.14.12268. Epub 2010 Jul 3. PMID: 20676040; PMCID: PMC3040960.

[2]Salton F, Volpe MC, Confalonieri M. EpithelialMesenchymal Transition in the Pathogenesis of Idiopathic Pulmonary Fibrosis. Medicina (Kaunas). 2019;55(4).

Point 4:In the JAK-STAT pathway, the activated STATs act as transcription factors which usually also activate the transcription of inhibitors of this pathway such as the SOCS or PIAS as a mechanism of feedback control. Did the authors test the effect of Baricitinib on the expression of those in either mRNA or protein levels?

Response 4:We thank the reviewers comment. JAK-STAT is a downstream signaling pathway regulated by TGF-β1, and recent researchs have shown that JAK-STAT pathway is abnormally activated during the process of PF[1]. It has been discovered that TGF-β1 was demonstrated to stimulate the Janus kinase/signal transducer and the activator of transcription 3 (STAT3) signaling axis and induce fibrosis. Meanwhile, other studies have shown that TGF-β-induced activation of STAT3 signaling could promote the activation of fibroblasts and tissue fibrosis[2,3]. The expression level of JAK and STAT3 is a crucial checkpoint for fibroblast activation and fibrosis [4] and baricitinib is a JAK1/2 inhibitor with strong anti-inflammatory activity and clinically used in the treatment of active rheumatoid arthritis and COVID-19 infection. So we established in vivo and in vitro models to further understand the anti-fibrogenic mechanism of baricitinib mainly by targeting JAK-STAT signaling pathway in pulmonary fibrosis. Based on our experimental results, we concluded that baricitinib could alleviate pulmonary fibrosis by inhibiting the activation of JAK1/2-STAT3 signaling pathways.

Whether baricitinib is involved in the feedback mechanism in the JAK-STAT pathway during pulmonary fibrosis and whether it affect expression of SOCS or PIAS in mRNA or protein levels has not been involved in our research, which provides us with ideas and directions for further research.

[1]Györfi AH, Matei AE, Distler JHW. Targeting TGF-β signaling for the treatment of fibrosis. Matrix Biol. 2018;68-69:8-27.

[2]Chakraborty D, Šumová B, Mallano T, et al. Activation of STAT3 integrates common profibrotic pathways to promote fibroblast activation and tissue fibrosis. Nat Commun. 2017;8(1):1130.

[3]Zehender A, Huang J, Györfi AH, et al. The tyrosine phosphatase SHP2 controls TGFβ-induced STAT3 signaling to regulate fibroblast activation and fibrosis. Nat Commun. 2018;9(1):3259.

[4]Chakraborty, D.; Šumová, B.; Mallano, T.; Chen, C.W.; Distler, A.; Bergmann, C.; Ludolph, I.; Horch, R.E.; Gelse, K.; Ramming, A.; et al. Activation of STAT3 integrates common profibrotic pathways to promote fibroblast activation and tissue fibrosis. Nat. Commun.

Point 5:The loading control GAPDH seems the same in both figures 8b and 9b. Please double check these. Also do the authors have better western blot images for p-JAK1, p-JAK2 and p-STAT3?

Response 5:We thank the reviewers comment. According to your suggestions, we have checked the gels used in the figures, the incorrectly placed loading control in Figure 8B has been corrected, and some western blot images including p-JAK1, p-JAK2 and p-STAT3 have been replaced to have a better reading experience. Thank you for your valuable suggestions.

Point 6:In figure 10, the authors depicted the mechanism of action of Baricitinib where they showed that the drug along with inactivation of JAK1/2-STAT3 pathway also blocks Akt and MAPK pathways. But there is no data shown in the manuscript in support of that.

Response 6:We thank the reviewers comment. Previous studies have confirmed that TGF-β1-Smad and non-Smad signaling pathways play important roles in regulating the proliferation and activation of pulmonary fibroblasts[1,2]. In the study, we used Mlg cells and NIH-3T3 cells lines to detect the effect of baricitinib on TGF-β1 signaling pathway, while the results indicated that there was no significant change in the Smad signaling activity after treated with baricitinib (Fig5 A-B in the manuscript). Meanwhile, we also detected the levels of proteins in non-Smad signaling pathways including phosphorylated P38, ERK, JNK and AKT in TGF-β1-treated Mlg cells and NIH-3T3 cells and the results revealed that baricitinib could restrain the activation of TGF-β1/non-Smad signaling pathway (Fig4 C-D in the manuscript).

According to your requirements, we have modified the description of the results, which is more friendly to the readers. Please check in the manuscript.

Reference:

[1]Finnson KW, Almadani Y, Philip A. Non-canonical (non-SMAD2/3) TGF-β signaling in fibrosis: Mechanisms and targets. Semin Cell Dev Biol. 2020;101:115-122.

[2]Walton KL, Johnson KE, Harrison CA. Targeting TGF-β Mediated SMAD Signaling for the Prevention of Fibrosis. Front Pharmacol. 2017;8:461.

Point 7:In lines 122-124, please mention clearly the equation and elaborate the method of calculation of the IHC staining densities.

Response 7:We thank the reviewers comment. We have supplemented relevant steps and calculation methods of the IHC staining densities and make the equation clearly.

First, change the IHC image into a grayscale image in ImageJ software, then convert the 8-bit grayscale image into an OD value. After selecting the parameters to be measured, select the positive signal area (red means selected), and finally get the positive integrated optical density value IOD (integrated optical density).

Minor observations:

Point 1:In line 103, please indicate the concentration of the HCl used.

Response 1:We thank the reviewers comment. We have indicated the concentration of the HCl used in the article, which is 6 mol/L.

Point 2:In line 254, figure legend of Figure 3, there is no panel called C or D. Please correct accordingly.

Response 2:We thank the reviewers comment. We have checked figure legend of Figure 3 and corrected the panel accordingly.

Point 3:There are some typos throughout the text. For example- in line 26 it should be attenuate, in line 51 (a vital role not an), in line 172 it should be horizontally, in line 307 (260 kD) etc.

Response 3:We thank the reviewers comment. We have checked the writing of the text and corrected typos in the manuscript. Thanks for your attentive suggestions.